# Genetic and dietary modulators of the inflammatory response in the gastrointestinal tract of the BXD mouse genetic reference population

Xiaoxu Li[1], Jean-David Morel[1], Giorgia Benegiamo[1], Johanne Poisson[1], Alexis Bachmann[1], Alexis Rapin[1], Jonathan Sulc[1], Evan Williams[2], Alessia Perino[3], Kristina Schoonjans[3], Maroun Bou Sleiman[1]*, Johan Auwerx[1]*

[1]Laboratory of Integrative Systems Physiology, Institute of Bioengineering, École Polytechnique Fédérale de Lausanne, Lausanne, Switzerland; [2]Luxembourg Centre for Systems Biomedicine, University of Luxembourg, Esch-sur-Alzette, Luxembourg; [3]Laboratory of Metabolic Signaling, Institute of Bioengineering, École Polytechnique Fédérale de Lausanne, Lausanne, Switzerland

**\*For correspondence:**
maroun.bousleiman@epfl.ch
(MBS);
admin.auwerx@epfl.ch (JA)

**Competing interest:** The authors declare that no competing interests exist.

**Abstract** Inflammatory gut disorders, including inflammatory bowel disease (IBD), can be impacted by dietary, environmental, and genetic factors. While the incidence of IBD is increasing worldwide, we still lack a complete understanding of the gene-by-environment interactions underlying inflammation and IBD. Here, we profiled the colon transcriptome of 52 BXD mouse strains fed with a chow or high-fat diet (HFD) and identified a subset of BXD strains that exhibit an IBD-like transcriptome signature on HFD, indicating that an interplay of genetics and diet can significantly affect intestinal inflammation. Using gene co-expression analyses, we identified modules that are enriched for IBD-dysregulated genes and found that these IBD-related modules share *cis*-regulatory elements that are responsive to the STAT2, SMAD3, and REL transcription factors. We used module quantitative trait locus analyses to identify genetic loci associated with the expression of these modules. Through a prioritization scheme involving systems genetics in the mouse and integration with external human datasets, we identified *Muc4* and *Epha6* as the top candidates mediating differences in HFD-driven intestinal inflammation. This work provides insights into the contribution of genetics and diet to IBD risk and identifies two candidate genes, *MUC4* and *EPHA6*, that may mediate IBD susceptibility in humans.

## eLife assessment

This **fundamental** study provides a framework for leveraging systems genetics data to dissect mechanisms of gut physiology. The authors provide **compelling** analyses to highlight diverse modes of interrogating intestinal inflammation, dietary response, and consequent impacts on inflammatory bowel disease. As a resource, it will have great utility for linking genetic variation and diet to gut-related pathophysiologies.

## Introduction

A long-term lipid-rich diet is associated with multiple metabolic disorders, such as obesity (*Hasegawa et al., 2020*), cardiovascular disease (*Lutsey et al., 2008*; *Maurya et al., 2023*), and systemic low-grade inflammation (*Duan et al., 2018*; *Christ et al., 2019*). The gastrointestinal tract is the primary

site of adaptation to dietary challenge due to its roles in nutrient absorption, immunity, and metabolism (*Enriquez et al., 2022*). Dietary challenges or other environmental or genetic factors can lead to prolonged inflammation and eventually damage the gastrointestinal tract (*Huang et al., 2017*; *Enriquez et al., 2022*). Inflammatory bowel disease (IBD) encompasses several chronic inflammatory gut disorders, including ulcerative colitis (UC) and Crohn's disease (CDs) (*Chang, 2020*; *Adolph et al., 2022*). Patients with IBD have a higher risk of developing colorectal cancer (CRC), one of the most lethal cancers (*Kim and Chang, 2014*; *Shah and Itzkowitz, 2022*). The incidence of IBD has increased worldwide (*Alatab et al., 2020*; *Freeman et al., 2021*) during the last decade in part due to increased consumption of lipid-rich diets (*Maconi et al., 2010*; *Hou et al., 2011*). Furthermore, mouse studies show that high-fat diet (HFD) leads to more inflammation in the dextran sulfate sodium (DSS)-induced UC models compared to normal diet (*Zhao et al., 2020*). However, the response to HFD is variable across individuals (*Zeevi et al., 2015*) and the association between the lipid-rich diet and the risk of IBD in clinical studies is inconclusive (*Kreuter, 2019*), possibly due to genetic factors underlying inter-individual variability in gut inflammation and dysbiosis (*Baumgart and Sandborn, 2012*). More than 200 risk genes associated with IBD were identified through human genome-wide association studies (GWAS) (*Huang et al., 2017*), which have implicated epithelial function, microbe sensing and restriction, and adaptive immune response as drivers (*Graham and Xavier, 2020*; *Kong et al., 2023*). However, there is still no effective treatment for IBD. Current therapies, such as anti-tumor necrosis factor alpha (TNF-α) antibodies (*Rutgeerts et al., 2005*) and integrin α4β7 antibodies, blocking leukocyte migration (*Feagan et al., 2013*), can temporarily alleviate inflammation in a subset of patients (*Rutgeerts et al., 2005*; *Feagan et al., 2013*) but cause adverse effects (*Harbord et al., 2017*) and fail to prevent relapses (*Doherty et al., 2018*). Therefore, it is important to understand the gene-by-environment (GxE) interactions underpinning preclinical gut inflammation that eventually evolves into IBD to aid in designing novel preventive and therapeutic strategies for intestinal inflammatory disorders.

Heterogeneity in clinical presentations as well as diversity in diet and lifestyle among human IBD patients renders human genetic studies challenging (*Molodecky et al., 2011*). Experiments in laboratory mice allow to control several environmental factors, such as temperature and diet, when exploring the genetic modulators of IBD and also enable the collection of several relevant tissues to help elucidate tissue-specific mechanisms (*Nadeau and Auwerx, 2019*; *Li and Auwerx, 2020*). In addition, to mirror the heterogeneity of human populations, genetically diverse populations, such as mouse genetic reference populations (GRPs), can be used in a systems genetics paradigm (*Nadeau and Auwerx, 2019*; *Li and Auwerx, 2020*). This not only allows the mapping of clinically relevant traits in controlled environments but also the characterization of intermediate molecular phenotypes from tissues that cannot easily be obtained in humans (*Williams et al., 2016*; *Li and Auwerx, 2020*). For example, studies on the molecular basis of non-alcoholic fatty liver disease in the Collaborative Cross founder strains illustrated the importance of the genetic background in determining susceptibility to steatosis, hepatic inflammation, and fibrosis (*Benegiamo et al., 2023*). Moreover, the BXD GRP was used to identify genetic variants associated with metabolic phenotype variation, such as bile acid homeostasis (*Li et al., 2022*), lipid metabolism in plasma (*Jha et al., 2018b*) and liver (*Jha et al., 2018a*), as well as mitochondrial dysregulation (*Williams et al., 2016*) using GWAS or quantitative trait locus (QTL) mapping (*Wu et al., 2014*; *Williams et al., 2016*). Thus, large mouse GRPs are useful tools for identifying the tissue-specific mechanisms of complex diseases.

In order to decipher the genetic and environmental contributions to the development of intestinal inflammation, we measured the colon transcriptome of 52 BXD strains fed with chow diet (CD) or HFD (*Williams et al., 2016*). HFD feeding from 8 to 29 wk of age induced an IBD-like transcriptomic signature in colons of some, but not all, BXD strains, uncovering a subset of BXD strains that could be susceptible to HFD-induced IBD-like state. Gene co-expression analysis revealed two IBD-related modules in the colons of HFD-fed mice, one of which is likely under the control of a module quantitative trait locus (ModQTL). Through a systems genetics prioritization of genes under this ModQTL, we identified candidate IBD-related genes that we validated using GWAS in the UK Biobank (UKBB) for human IBD.

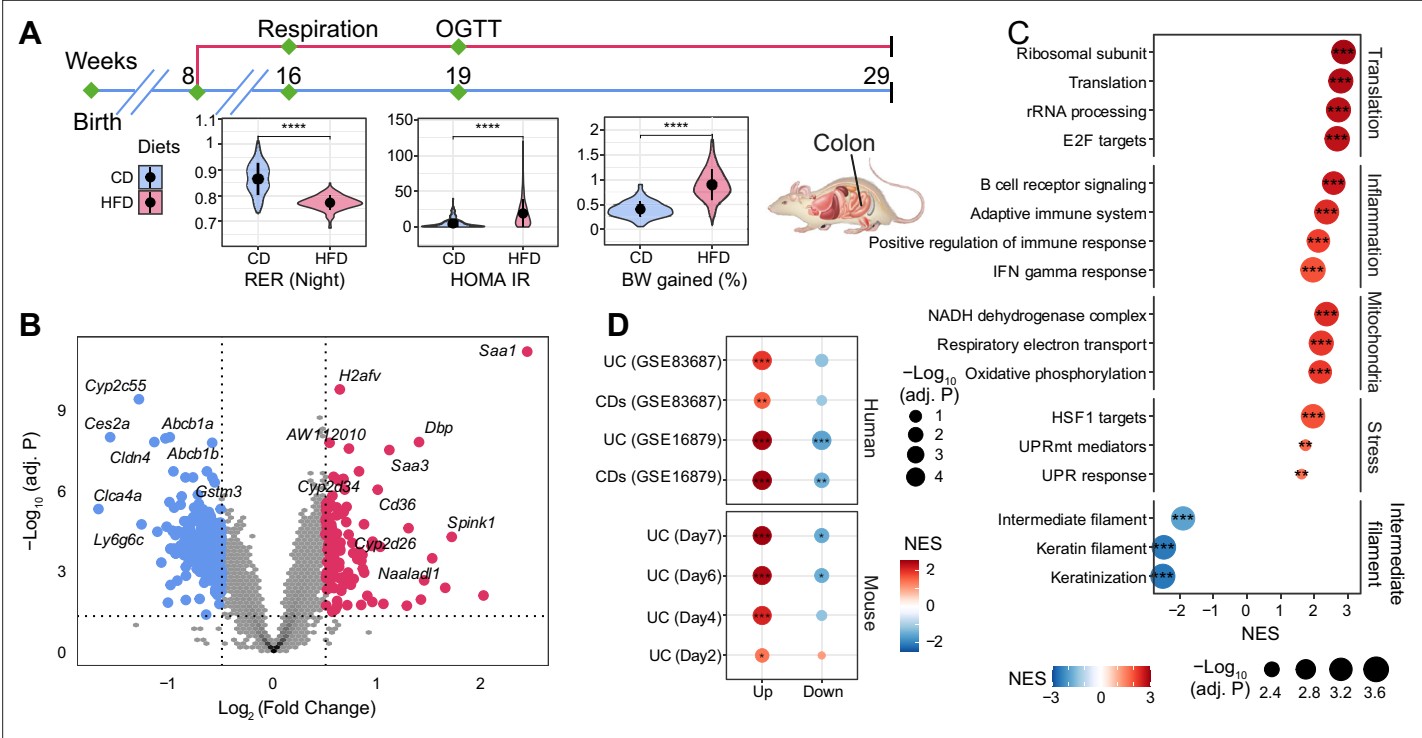

**Figure 1.** The effect of long-term high-fat diet (HFD) on gene expression in BXD colons. (**A**) Graphical representation of a pipeline of a previously described BXD mouse study (*Williams et al., 2016*). Mice were fed HFD or chow diet (CD) starting from 8 wk of age, and metabolic phenotyping was performed as indicated. Mice were sacrificed at 29 wk of age, and multiple organs were collected and frozen for further analyses. BXD colon transcriptomes were analyzed in this study. CD is indicated in blue, and HFD is indicated in red. p-Values were calculated by two-tailed Student's *t*-test and indicated as follows: *p<0.05; **p<0.01; ***p<0.001; ****p<0.0001. (**B**) Volcano plot showing the HFD effect on BXD colon transcriptomes compared to CD and the up- and downregulated differentially expressed genes (DEGs, absolute log₂(fold change) > 0.5 and Benjamini–Hochberg [BH]-adjusted p-value (adj. p<0.05)) are highlighted in red and blue, respectively. (**C**) Gene set enrichment analysis showing the effect of HFD on gene expression in BXD colons. Gene sets were grouped into five categories: translation, inflammation, mitochondria, stress, and intermediate filament. Normalized enrichment scores (NES) are represented by color, and -Log₁₀ (BH-adj. P) are represented by dot size and indicated as follows: *adjusted p-value<0.05; **adjusted p-value<0.01; ***adjusted p-value<0.001. (**D**) Enrichment analysis of molecular signatures of mouse and human inflammatory bowel disease (IBD) on the transcriptome of BXD colons. UC: ulcerative colitis; CDs: Crohn's disease.

The online version of this article includes the following figure supplement(s) for figure 1:

**Figure supplement 1.** Transcriptome profile in BXD colons.

## Results

### HFD feeding leads to highly variable transcriptomic adaptations in the colon of BXD strains

For this study, we used an extensively characterized BXD mouse panel of 52 BXD strains fed with a CD or HFD from 8 to 29 wk of age (*Williams et al., 2016*; *Jha et al., 2018a*; *Jha et al., 2018b*), in which we mapped genetic determinants of metabolic traits in the liver (*Williams et al., 2016*; *Jha et al., 2018a*) and plasma (*Jha et al., 2018b*). These mice underwent metabolic phenotyping, with many metabolic traits being altered by HFD (*Figure 1A*), and multiple organs were harvested and flash-frozen for future use (*Williams et al., 2016*). Here, we focused on proximal colon samples from this population and performed microarray-based transcriptome analysis of this tissue (*Figure 1A*).

Principal component analysis (PCA) of all transcriptomes (*Figure 1—figure supplement 1A*) showed the first principal component (PC1) separated mice by diet, indicative of a global diet effect in the population. Nevertheless, transcriptomes of several strains (such as BXD12, BXD84, and BXD81) on HFD had very similar PC1 values to their CD counterparts (*Figure 1—figure supplement 1B*), suggesting that they were resistant to dietary changes. Similarly, BXD strains did not cluster completely by diet based on hierarchical clustering analysis indicating that the genetic differences can override the impact of diet on the transcriptome in the colon (*Figure 1—figure supplement 1C*).

To obtain a global, strain-independent, view of the HFD effect, we performed a differential expression analysis and identified 115 up- and 295 downregulated differentially expressed genes (DEGs, absolute log$_2$(fold change) > 0.5 and Benjamini–Hochberg [BH]-adjusted p-value<0.05, *Figure 1B*). Of note, *Cldn4*, one of claudins implicated in intestinal permeability (*Ahmad et al., 2017*), was significantly downregulated and serum amyloid A (*Saa1* and *Saa3*), which have been involved in the inflammatory response (*Ye and Sun, 2015*; *Tannock et al., 2018*), were upregulated upon HFD (*Figure 1B*). Furthermore, gene set enrichment analysis (GSEA) showed an upregulation of inflammation, cell proliferation and translation, mitochondrial respiration, and stress response-related pathways upon HFD, while genes involved in the intermediate filament – that contribute to maintaining intestinal barriers (*Misiorek et al., 2016*; *Mun et al., 2022*) – were downregulated (*Figure 1C*). All in all, the transcriptome data are consistent with an HFD-induced downregulation of components of the intestinal barrier, enhanced permeability, induction of the unfolded protein response (UPR) and increased inflammation in BXD colons, much like HFD does in humans (*Bischoff et al., 2014*). However, as in humans, not every strain exhibited the same response to dietary challenges. GSEA applied individually to the diet effect in each strain showed a high degree of diversity in the inflammatory response (*Figure 1—figure supplement 1D*). For example, BXD44, 45, and 55, highlighted in red, were the three most susceptible strains to gut inflammation upon HFD, whereas BXD1, 67, and 85, colored in green, showed no significant enrichment in gut inflammation. This diversity in responses provided the basis for a systems genetics investigation of HFD-driven gut inflammation determinants in the BXD.

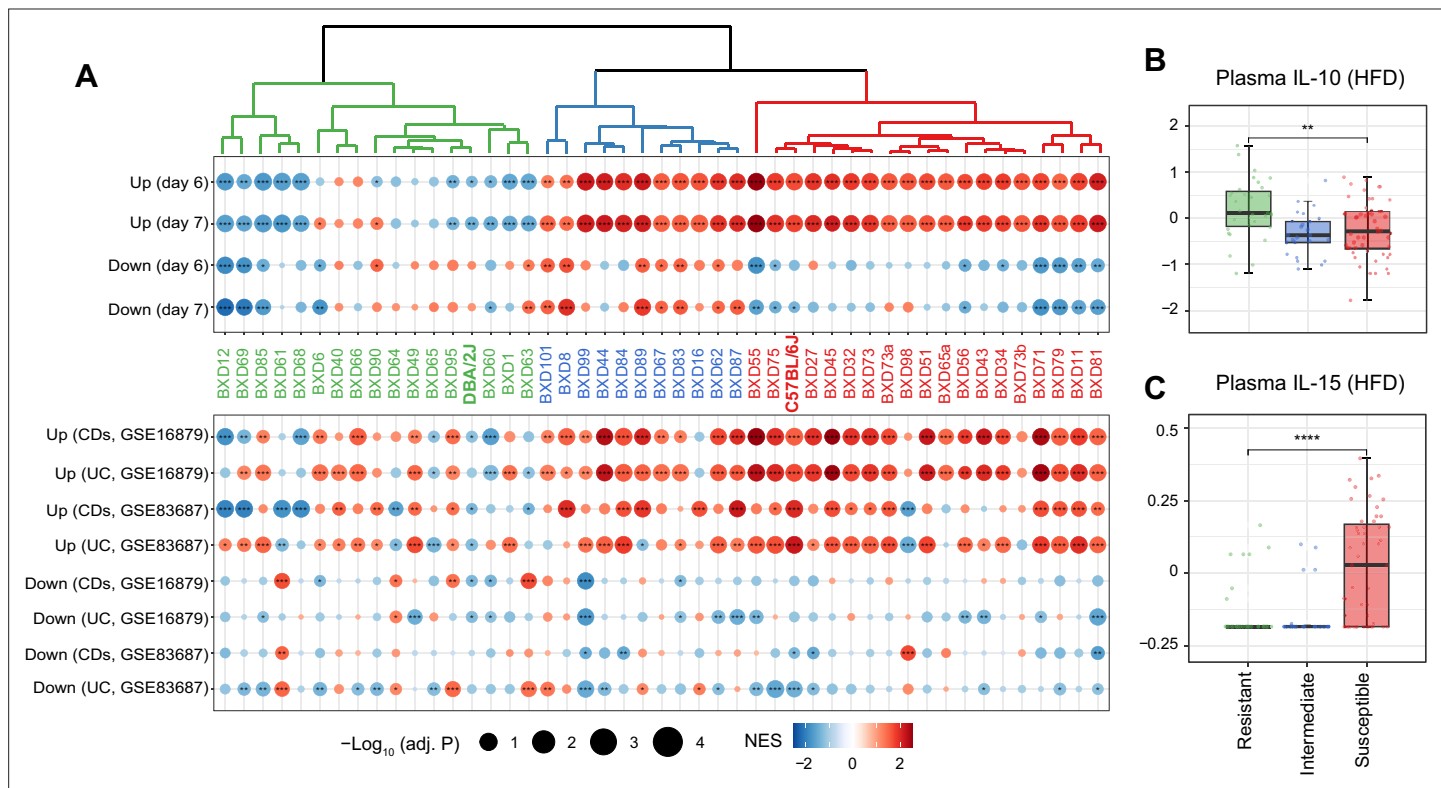

**Figure 2.** Identifying susceptible strains to high-fat diet (HFD)-induced inflammatory bowel disease (IBD)-like inflammation and their effect on plasma cytokines. (**A**) Enrichment analysis of the molecular signatures in human (bottom panel) and mouse IBD (top panel) models on the gene expression of individual BXD colons. Dendrogram showing the propensity to develop a mouse IBD-like signature among BXD strains upon HFD. BXD strains were divided into three clusters: susceptible (in red), intermediate (in blue), and resistant strains (in green). Normalized enrichment scores (NES) are represented by color, and -Log$_{10}$(BH-adjusted p-values) are represented by dot size and indicated as follows: *adjusted p-value<0.05; **adjusted p-value<0.01; ***adjusted p-value<0.001. UC: ulcerative colitis; CDs: Crohn's disease. (**B, C**) Boxplots showing the effect of susceptible strains on plasma IL-10 (**B**) and IL-15 (**C**) level compared to resistant strains. p-Values were calculated by two-tailed Student's *t*-test and indicated as follows: *p<0.05; **p<0.01; ***p<0.001; ****p<0.0001.

## The transcriptomic response to HFD of a subset of BXD strains resembles DSS-induced UC

IBD is characterized by increasing inflammation in the gastrointestinal tract (*Adolph et al., 2022*). To investigate the disease relevance of the chronic inflammation seen in BXD colons upon HFD, we extracted the transcriptomic signatures from DSS-induced mouse UC models (*Czarnewski et al., 2019*) and two IBD human studies (GSE16879 [*Arijs et al., 2009*] and GSE83687 [*Peters et al., 2017*], 'Materials and methods') and used these signatures as custom gene sets in GSEA on the global HFD effect. DSS is widely used to induce UC in mouse models and disease severity increases over time (*Czarnewski et al., 2019*). GSEA showed that DSS-induced genes from days 4 (early inflammatory phase), 6 and 7 (acute inflammatory phase) were significantly enriched in genes upregulated by HFD, especially the dysregulated genes in the later stage of DSS-induced UC (*Figure 1D*, bottom panel). Similarly, genes involved in human IBD (UC and CDs) were also enriched in those same genes (*Figure 1D*, top panel). The same trend was observed for downregulated genes in mouse and human IBD, which were negatively enriched (*Figure 1D*), illustrating that HFD induced an IBD-like transcriptomic signature in BXD colons.

While the average response across all BXDs shared features of mouse and human IBD, we assessed the strain specificity of this response by measuring each strain's response to IBD using GSEA (*Figure 2A*). Hierarchical clustering of the normalized enrichment scores (NES) in mouse IBD datasets classified the BXDs into three groups: susceptible strains highlighted in red (19 strains), intermediate strains represented in blue (11 strains), and resistant strains colored in green (17 strains) (*Figure 2A*, top panel). Of note, in line with colon histological lesions comparison of DSS-induced colitis mouse models in the literature (*Mähler et al., 1998*), the C57BL/6J strain, one of the parental strains of the BXDs, was classified as one of the susceptible strains while the other parental strain DBA/2J belonged to the resistant group (*Figure 2A*, top panel), suggesting that genetic determinants inherited from the parental strains may determine the susceptibility of BXD strains to HFD-induced IBD-like inflammation in the colons.

To establish the functional relevance of this transcriptome-based classification on systemic inflammation, we compared plasma cytokine levels of these three groups under HFD (*Williams et al., 2016*). Interestingly, the susceptible group has significantly lower levels of the anti-inflammatory cytokine–interleukin (IL)-10 (*Figure 2B*, two-tailed *t*-test p<0.01) and increased the proinflammatory cytokine–IL-15 (*Figure 2C*, two-tailed *t*-test p<0.0001) compared to the resistant strains. *IL10* itself has been identified as an IBD-related candidate gene using GWAS in humans (*Franke et al., 2008*) and IL-10-deficient mice are also well-known mouse model for IBD research (*Keubler et al., 2015*). IL-15 is another important cytokine involved in intestinal inflammation and is elevated in the human guts with IBD (*Liu et al., 2000*). IL-15 knockout mice are also reported to have less severe symptoms, such as weight loss and histological scores, following DSS administration (*Yoshihara et al., 2006*). In summary, susceptibility to HFD-induced IBD-like inflammation in the colon, as assessed by changes in levels of genes associated with IBD, correlates with markers of the general inflammatory status of mice.

## Identifying IBD-related gene modules in BXD colons

Since different BXD strains seem to exhibit different susceptibility to IBD, we set out to explore gene expression signatures underlying these differences. For that, we used Weighted Gene Co-expression Analysis (WGCNA) to construct CD- and HFD-specific gene co-expression networks to identify modules of co-expressed genes (*Figure 3A*, *Supplementary file 1*). Disease-associated modules were then defined as modules under HFD are significantly enriched in mouse DSS-induced UC signatures by an over-representation analysis (ORA, BH-adjusted p-value<0.05 and number of enriched genes >5, *Figure 3A*). The HFD co-expression network consisted of 39 modules ranging in size from 34 to 1853 genes and containing a total of 14,723 genes (*Supplementary file 1*). We visualized this network using Uniform Manifold Approximation and Projection (UMAP) (*Figure 3B*), reflecting that the majority modules were closely connected in the co-expression network.

Enrichment analyses indicated that modules HFD_M9 (484 genes), HFD_M16 (328 genes), and HFD_M28 (123 genes) were enriched with genes that are upregulated by DSS-induced colitis, while HFD_M15 (368 genes), HFD_M24 (159 genes), and HFD_M26 (135 genes) were significantly enriched with downregulated genes (*Figure 3C*). Of note, more than 20% of the genes involved in HFD_M9 and HFD_M28 were part of the dysregulated genes of the acute phase of mouse UC (days 6 and 7)

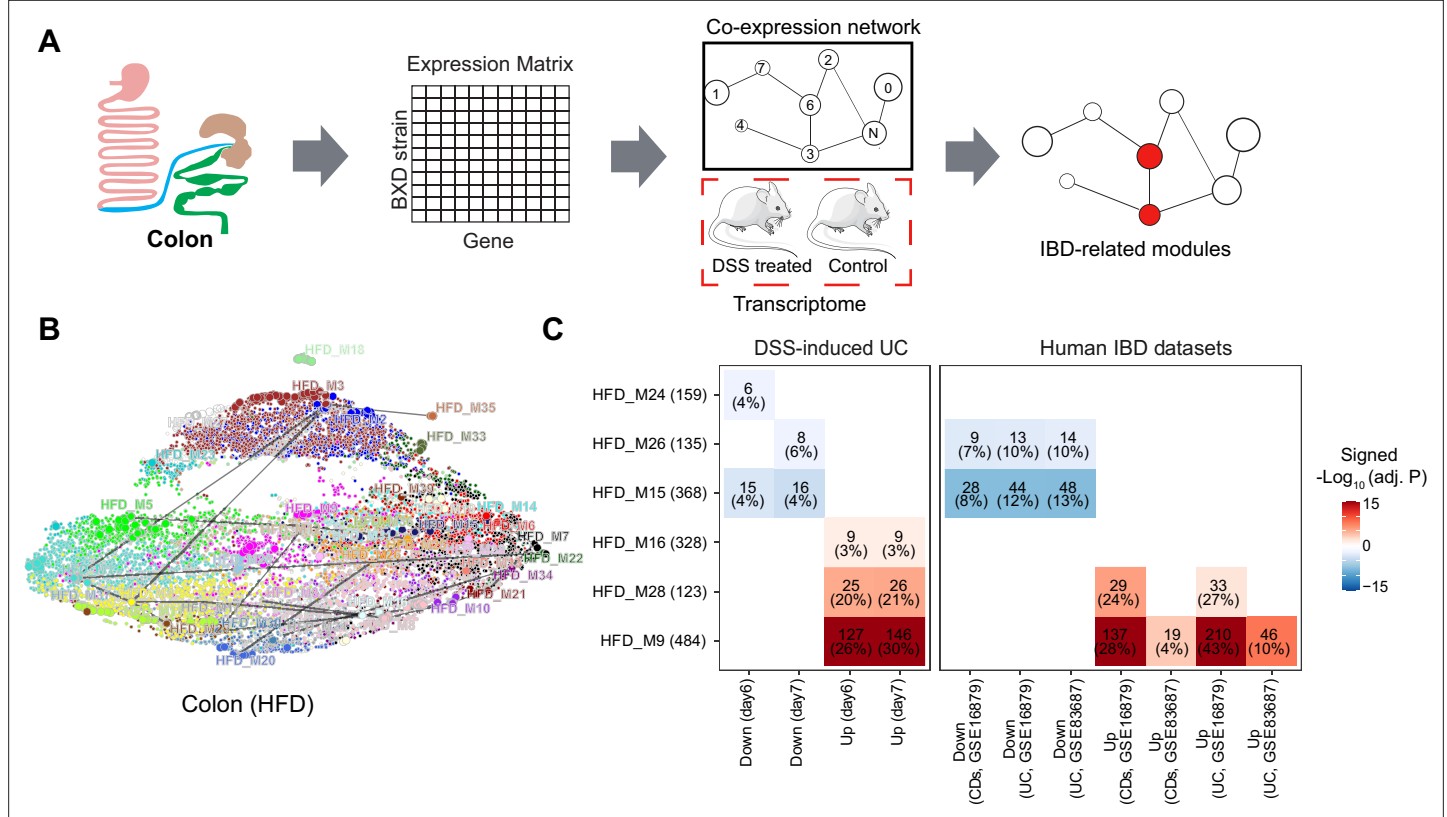

**Figure 3.** Identifying inflammatory bowel disease (IBD)-related gene modules. (**A**) Pipeline for exploring the IBD-associated gene modules. A co-expression gene network was constructed based on the transcriptome of BXD colons under high-fat diet (HFD). IBD-associated modules were then defined as gene modules under HFD that are significantly clustered in mouse dextran sulfate sodium (DSS)-induced ulcerative colitis (UC) signatures. (**B**) Uniform Manifold Approximation and Projection (UMAP) representation of the co-expression gene network under HFD. Thirty-nine co-expression modules are represented in the corresponding color, and the correlated modules (Spearman correlation coefficient between the eigengene of modules >0.7) are linked by a gray line. (**C**) Heatmap showing the enrichment of co-expression modules in mouse and human IBD gene signatures, and the number and percentage of enriched genes are labeled. The number of enriched genes divided by the number of genes involved in the respective module is defined as the percentage of enriched genes. Signed –Log$_{10}$(adj. p) indicates Benjamini–Hochberg (BH)-adjusted p-value and the enriched gene set. Enriched modules of up- and downregulated genes upon disease are highlighted in red and blue, respectively. CDs: Crohn's disease.

The online version of this article includes the following figure supplement(s) for figure 3:

**Figure supplement 1.** Exploring inflammatory bowel disease (IBD)-associated co-expression modules under chow diet (CD).

(**Figure 3C**). Interestingly, genes perturbed during IBD pathogenesis in humans were also enriched in HFD_M9 and HFD_M28 (**Figure 3C**).

While IBD-related genes were predominantly found in HFD modules, we also found that two modules, CD_M28 (185 genes) and CD_M32 (142 genes), in CD-fed mouse colons were associated with IBD (**Figure 3—figure supplement 1A**). These two modules significantly overlapped with the IBD-related HFD_M9 and HFD_M28 modules, respectively (BH-adjusted p-value<0.05) (**Figure 3— figure supplement 1B**). Moreover, the molecular signatures underlying human UC and CDs were also clustered in these two modules (CD_M28 and CD_M32) under CD (**Figure 3—figure supplement 1C**). Collectively, the co-expression and enrichment analyses identify HFD_M9 and HFD_M28 as IBD-related modules on which we focus our subsequent investigation.

## Identifying biological roles and transcriptional regulation of the IBD-related modules

To identify the biological function of the IBD-related modules, we performed enrichment analyses using the Hallmark database and the cell-type gene signatures (**Kong et al., 2023**; 'Materials and methods'). Genes in HFD_M9 were enriched in KRAS signaling and inflammation-related pathways, while HFD_M28 was enriched in IFN-α/γ responses (BH-adjusted p-value<0.05) (**Figure 4A**). Both modules were

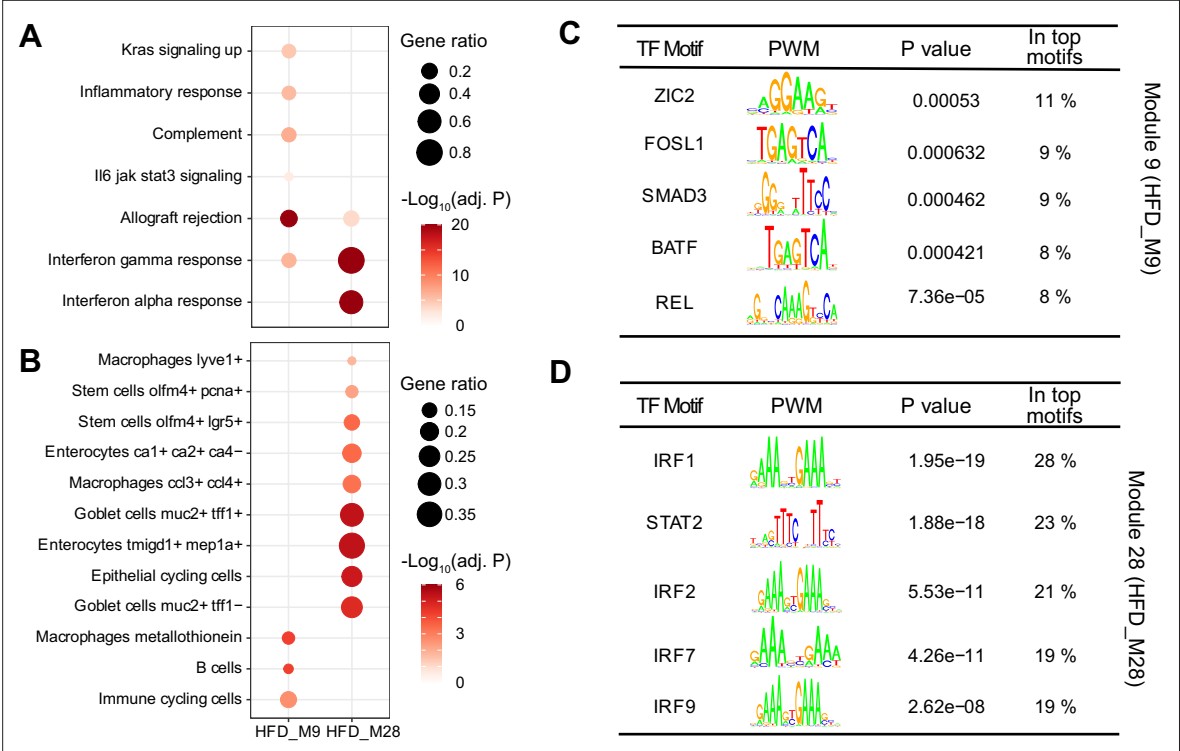

**Figure 4.** Biological interrogation of identified inflammatory bowel disease (IBD)-related modules. (**A, B**) Dotplots showing the enrichment of IBD-related modules in hallmark genesets (**A**) and cell-type gene signatures of inflamed colon in Crohn's disease patients (**B**). Gene ratios higher than 0.1 are shown and represented by dot size. Dots are colored by -log$_{10}$(BH-adjusted p-values). (**C, D**) The enriched motifs for promoters of the genes involved in module HFD_M9 (**C**) and HFD_M28 (**D**). The significantly enriched motifs (p-value<0.001) were ranked based on the percentage of enriched promoters (In top motifs) and then the top five TFs were selected. TF: transcription factor; PWM: positional weight matrix.

enriched in IFN-γ response genes (*Figure 4A*). IFN-γ is an essential cytokine for innate and adaptive intestinal immune responses (*Brasseit et al., 2018*). It has been reported to play a key role in mouse (*Ito et al., 2006*) and human (*Tilg et al., 2002*) IBD pathogenesis, and was identified as a potential therapeutic target to alleviate inflammatory response in IBD (*Li et al., 2021*). In addition, genes that are dysregulated in immune cells of CDs patients (macrophages, B cell, and immune cycling cells) were enriched in HFD_M9 (*Figure 4B*). In contrast, genes of HFD_M28 were not only enriched for genes that are dysregulated in immune cells, but also in intestinal epithelial cells of diseased individuals, such as Goblet and stem cells (*Figure 4B*). Overall, HFD_M9 and HFD_M28 are both involved in inflammatory response, while genes involved in HFD_M28 also potentially influence intestinal epithelial barrier.

To identify transcriptional drivers of the two IBD-related modules, we performed a transcription factor (TF) enrichment analysis ('Materials and methods') and found that ZIC2, SMAD3, REL, FOSL1, and BATF are the top enriched TFs for the genes in HFD_M9 (*Figure 4C*), while the expression of genes in module HFD_M28 may be regulated by interferon regulatory factors (IRFs, IRF1, IRF2, IRF7, and IRF9) and the signal transducer and activator of transcription families (STAT, STAT2) (*Figure 4D*). In fact, most of these TFs have been reported to be involved in gut inflammation. For example, *Smad3* mutant mice were more susceptible to intestinal inflammation (*Yang et al., 1999*). Moreover, the IFN-STAT axis is essential to initiate the type I IFN induction that is critical for human immune defense, such as IBD diseases (*Stolzer et al., 2021*) and primary immunodeficiency diseases (*Mogensen, 2018*), as well as for disease tolerance (*Mottis et al., 2022*). Collectively, we have identified TFs that likely control the expression of the two IBD-related modules to play an essential role in gut inflammation regulation.

## Identifying ModQTLs for IBD-related modules and filtering of candidate genes

To analyze how the genotype impacts the IBD-like inflammatory response associated to HFD, we performed ModQTL mapping analysis for both IBD-related modules (HFD_M9 and HFD_28) (*Figure 5A*). We found a suggestive QTL for HFD_M28 (p-value<0.1) on chromosome 16 containing 552 protein-coding genes (*Figure 5A*, *Supplementary file 2*). The ModQTL analysis was also performed on the modules that are significantly enriched in IBD-downregulated genes (HFD_M15, HFD_M24, and HFD_M26), but no significant or suggestive QTLs were detected. Therefore, we focused on the QTL for IBD-induced genes in HFD_M28 and annotated its candidate genes based on three criteria (*Figure 5B*): (1) presence of high-impact genetic variants (such as missense and frameshift variants) in BXDs, (2) association with inflammation based on literature mining ('Materials and methods'), and (3) presence of *cis*-expression QTLs (eQTLs), that is, whether the expression of the gene is controlled by the QTL. The 27 genes satisfying at least two of the above criteria were considered as candidate genes driving the expression of module HFD_M28 (*Figure 5C*).

To further prioritize candidate genes regulating module HFD_M28, we applied GWAS to detect CDs- and UC-associated genetic variants using whole-genome sequence (WGS) dataset in UKBB (*Figure 5C*, 'Materials and methods'). Interestingly, the genetic variants of two genes under the QTL peak, that is, *ephrin type A receptor 6* (*EPHA6*, p-value=2.3E-06) (*Figure 5C*, *Figure 5—figure supplement 1A*, *Supplementary file 3*) and *Mucin 4* (*MUC4*, p-value=1.2E-06) (*Figure 5C*, *Figure 5—figure supplement 1B*, *Supplementary file 4*) were also associated with UC in humans. *EPHA6* belongs to Eph/Ephrin Signaling, and this pathway has been associated with gut inflammation (*Coulthard et al., 2012*) and proposed as a potential target to alleviate the inflammatory response in IBD (*Grandi et al., 2019*), but the association between *EPHA6* and IBD is not explored yet. The Gene-Module Association Determination (G-MAD) (*Li et al., 2019*; https://systems-genetics.org/gmad) also revealed that expression of *Epha6* in mouse gastrointestinal tract correlates with genes involved in inflammation-related pathways, such as IL-6 production and regulation of inflammatory response (*Figure 5D*, *Supplementary file 5*). *MUC4* is a transmembrane mucin (*Gao et al., 2021*) and highly expressed in gastrointestinal tract according to the human protein atlas (*Uhlén et al., 2015*; https://www.protein-atlas.org/humanproteome/tissue/intestine; *Figure 5C*). The expression of *MUC4* in the human gastrointestinal tract correlates with genes that are enriched for CRC and O-linked glycosylation based on G-MAD (*Li et al., 2019*; *Figure 5E*, *Supplementary file 5*). O-linked glycans are expressed by the intestinal epithelium to maintain barrier function, especially mucin type O-glycans, and gut disorders can be affected by dysfunction of O-linked glycosylation (*Brazil and Parkos, 2022*). Moreover, *MUC4* is upregulated in enterocytes and Goblet cells in colons of CDs patients (*Figure 5F*). *MUC4* hence is a strong candidate because of its role in maintaining the intestinal epithelium and controlling the gut inflammatory response (*McGuckin et al., 2011*) and *EPHA6* might be a novel candidate gene to impact gut inflammation. Based on the results of our QTL mapping, human GWAS in UKBB, and existing literature, we hypothesize that *MUC4* and *EPHA6* impact on colon integrity and inflammation and may be important players in gut inflammation or IBD triggered by an unhealthy, lipid-rich diet.

However, it is unclear through what mechanisms the genetic variants in the candidate genes affect IBD susceptibility. One possibility is that genetic variation leads to altered levels of expression of the gene, ultimately affecting disease susceptibility. To test this possibility, we examined the GTEx resource (*Lonsdale et al., 2013*) and found that *MUC4*, but not *EPHA6*, has *cis*-eQTLs in the sigmoid and transverse colon. To establish likely causal links with IBD incidence, we used these associations as instruments in a two-sample Mendelian randomization (MR) (*Hemani et al., 2017*; *Hemani et al., 2018*) analysis. Using publicly available GWAS summary statistics for IBD, CDs, and UC (*Liu et al., 2015*; *Elsworth et al., 2020*) as outcomes, we found suggestive evidence that increased expression of *MUC4* in the sigmoid, but not transverse, colon may increase the risk of IBD (nominal p-value=0.033, *Supplementary file 6*). No eQTLs were reported for *EPHA6* in the colon, precluding us from investigating the potential consequences of changes in its expression in these tissues.

## Discussion

Dietary, environmental and genetic factors have all been reported to influence intestinal inflammation (*Adolph et al., 2022*). Indeed, HFD can impair the intestinal epithelial barrier and trigger preclinical

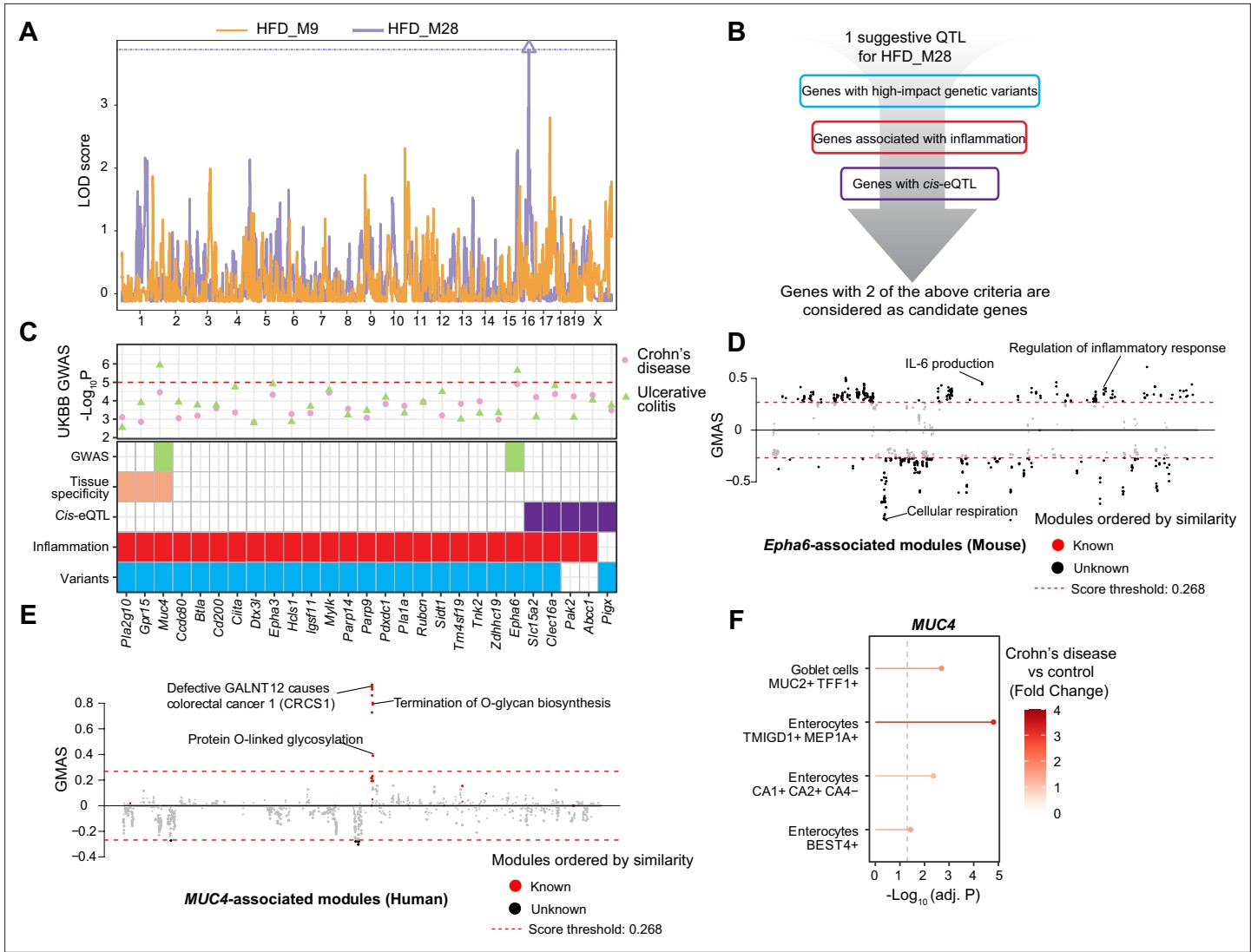

**Figure 5.** Module quantitative trait locus (ModQTL) mapping for two inflammatory bowel disease (IBD)-related modules and the prioritization of candidate genes. (**A**) Manhattan plot showing the ModQTL mapping result for disease-related modules HFD_M9 and HFD_M28. ModQTL maps of HFD_M9 and HFD_M28 are indicated in orange and purple, respectively. The threshold calculated by permutation test (p<0.1) for HFD_M28 is represented by a purple dashed line. (**B**) The filtering criteria for selecting candidate genes under the ModQTL peak for HFD_M28. Genes with two of the described criteria are considered as candidate genes. (**C**) The most significant associations between 27 candidate genes under the ModQTL peak and Crohn's disease or ulcerative colitis (UC) identified through genome-wide association studies (GWAS) according to whole-genome sequence in the human UK Biobank (UKBB) are shown in the scatter plot (top panel). Crohn's disease and UC are indicated by pink circle and green triangle, respectively. The threshold (-Log$_{10}$ p-value=5) is represented by a red dashed line. Heatmap showing the identified 27 candidate genes of module HFD_M28 (bottom panel). Variants colored in blue indicate genes with high-impact genetic variants in BXD mice (including missense, frameshift, initiator codon, splice donor, splice acceptor, in-frame deletion, in-frame insertion, stop lost, stop gained). Inflammation is indicated in red and represents genes associated with inflammation based on literature mining. *Cis*-eQTL colored in purple indicates genes with *Cis*-eQTLs. Tissue specificity colored in orange means genes that are highly expressed in human intestine (data were downloaded from human protein atlas, https://www.proteinatlas.org/humanproteome/tissue/intestine [***Uhlén et al., 2015***]). GWAS result of UC in UKBB colored in green indicates that genes are significantly associated with human UC. (**D, E**) Manhattan plots showing the associated gene expression modules of *Epha6* in mouse gastrointestinal tract (**D**) and that of *MUC4* in human gastrointestinal tract (**E**) (data from https://systems-genetics.org/gmad [***Li et al., 2019***]). The threshold is represented by the red dashed line (absolute Gene-Module Association Score [GMAS] ≥ 0.268). Terms above the threshold are identified as the significant associated terms. GO terms or gene modules are ranked by similarity. Known associated terms are shown as red dots, and new significant associated terms are colored in black. (**F**) Dot plot showing that the expression of *MUC4* was higher in four cell types of human inflamed colon with Crohn's disease.

The online version of this article includes the following figure supplement(s) for figure 5:

**Figure supplement 1.** Genome-wide association studies (GWAS) for ulcerative colitis (UC) and Crohn's disease (CDs) in humans.

inflammation in the gastrointestinal tract, eventually leading to inflammatory disorders of the gut (*Enriquez et al., 2022*). In addition, genetic factors identified by GWAS can also predispose to IBD. For example, the interleukin-1 and -7 receptors (*IL1R2* and *IL7R*) were identified as candidate genes that regulate the immune response in IBD (*Khor et al., 2011*). However, the heterogeneity of diet and other environmental factors in human studies limits our ability to identify GxE interactions and pinpoint the genes and pathways involved in diet-induced gut inflammation. Studies in model organisms such as the mouse, where the environment can be carefully controlled, provide a valuable complement to human genetics studies that by nature are mainly observational (*Nadeau and Auwerx, 2019*; *Li and Auwerx, 2020*). Unfortunately, most mouse studies only evaluate mice from a single genetic background, limiting their generalizability and translatability to humans (*Nadeau and Auwerx, 2019*; *Li and Auwerx, 2020*). Conversely, GRPs such as the BXDs can mimic at least in part the heterogeneity of human populations and allow us to estimate the effect of GxE interactions on complex diseases (*Jha et al., 2018a*; *Li and Auwerx, 2020*).

Here, we utilized a panel of 52 BXD genetically diverse mouse strains fed with either HFD or CD to explore the genetic and dietary modulators of inflammation seen in the colon transcriptomes using systems genetics approaches. The colon transcriptomic response to HFD in this mouse population recapitulated several of the general features observed in DSS-induced UC mouse models and human IBD patients. In particular, we identified the upregulation of inflammation-related genes and the UPR as well as the downregulation of intercellular adhesion-related genesets as common signatures induced by HFD (*Kreuter, 2019*). Moreover, our dataset not only was informative about the transcript changes of IBD at the population level, but also unveiled extensive strain-specific effects that allowed us to classify strains based on their propensity to develop IBD-like signatures. The fact that these susceptibility groups also differed in anti- and pro-inflammatory plasma cytokine levels (IL-10 and IL-15, respectively) suggests a relation between these tissue-specific transcriptional signatures and systemic low-grade inflammation. Since gene interactions determine cellular processes and the molecular functions of correlated genes are often similar (*Nayak et al., 2009*), we attempted to elucidate the mechanism underlying the diversity of IBD-like signatures and chronic inflammation in BXD colons using gene co-expression analyses. This led us to identify two IBD-related gene modules (HFD_M9 and HFD_M28).

As most differentially expressed genes are likely to be driven by and not be a cause of disease (*Porcu et al., 2021*), we attempted to understand whether the signatures in the colon are causes or consequences of chronic inflammation. A first step was to characterize possible transcriptional and genetic regulators of IBD-related modules. Enrichment analyses showed that both IBD-associated modules largely consisted of immune response-related genes. Specifically, genes involved in HFD_M9 and HFD_M28 are both differentially expressed in immune cells in inflamed tissues of CDs patients (*Kong et al., 2023*). Moreover, the HFD_M28 module was enriched for TF motifs of STAT2 and IRF family, and HFD_M9 for SMAD3 and REL, which were illustrated to control the expression of these gut inflammation-related genes, and influence the inflammatory response triggered by HFD in the colon.

While we found IBD-related gene modules and the TFs driving their expression, the genetic drivers of the diversity of gut inflammatory responses observed across the BXDs remained elusive. To find candidate genes causing gut inflammation upon HFD, we then performed Module QTL (ModQTL) analysis and allocated a suggestive ModQTL that may be controlling one of IBD-related module (HFD_M28) under HFD. Importantly, through our prioritization scheme for the genes under the ModQTL, we identify two plausible candidates, *Epha6* and *Muc4*, that have high-impact variants in the BXDs, are related to inflammation, and harbor variants in humans that are associated with IBD based on UKBB GWAS result. Mendelian randomization analysis suggests that higher expression of *MUC4* in the sigmoid colon may increase the risk of IBD. Furthermore, Muc4 knockout mice have been shown to be more resistant to DSS-induced UC through upregulating the expression of *Muc2* (mucin secretion) and *Muc3* (transmembrane mucin) (*Das et al., 2016*). A GWAS also indicated that mutations in *EPHA6* increase risk for CRC (*Guda et al., 2015*), but its potential association with IBD is a new finding. Therefore, these results point to important potential roles of *Muc4* and *Epha6* in gut chronic inflammation leading to inflammatory gut disorders.

Although studies in the BXD cohort are limited to variants present in the parental strains, C57BL/6J and DBA/2J, our analysis nevertheless shows how genetic diversity in this population allows us to detect the genetic modulators of chronic intestinal inflammation, that are more difficult to identify

in widely used IBD mouse models on a single genetic background. In support of the generalizability of our data, the identified candidate genes in our mouse models were also associated to human UC, demonstrating that chronic inflammation induced upon HFD feeding may indeed be a prelude to human UC.

In conclusion, our systems genetics investigation of the colon in a controlled GRP, complemented with human GWAS studies, enabled the prioritization of modulators of IBD susceptibility that were generalizable to the human situation and may have clinical value.

# Materials and methods
## Population handling
Mice were studied as previously described (*Williams et al., 2016*) and multiple organs were harvested for further analysis. Briefly, in groups of 3–5 animals from the same strain and diet, in isolator cages with individual air filtration (500 cm², GM500, Tecniplast) and provided water ad libitum. Mice were fed CD ad libitum until 8 wk of age. From 8 wk to 29 wk, half of the cohort was fed ad libitum HFD and the rest continued to be fed a CD (*Figure 1A*). CD composition: 18% kcal fat, 24% kcal protein, and 58% kcal of carbohydrates (Teklad Global 18% Protein Rodent Diet 2018 chow diet, Envigo, Indianapolis, USA). HFD composition: 60.3% kcal fat, 18.4% kcal protein, and 27.3% kcal of carbohydrates (Teklad Custom Diet TD.06414, Envigo). All mice were fasted overnight (from 6 pm to 9 am) prior to euthanasia. All procedures were approved by the veterinary office of canton Vaud under animal experimentation license number VD2257. In this work, proximal colons were extracted from the bio-banked samples and we did not use any new animals.

## Transcriptome of the proximal colon in BXDs
An ~1 cm portion of the proximal half part of the colon was excised following euthanasia, washed in PBS, and immediately stored in liquid nitrogen. Approximately five animals of the same strain fed the same diet were pooled at equal mass concentration for further RNA extraction. Total RNA was extracted using Directzol (Zymo Research) including the DNase digestion step. Then, 100 ng of total RNA was amplified using the Ambion WT Expression Kit from Life Technologies (part number 4411974) and 5500 ng of cDNA was fragmented and labeled using the Affymetrix WT terminal labeling kit (part number 900671) all following the manufacturer's protocols. Labeled cDNA was hybridized on an Affymetrix Clariom S Assay microarray platform (GPL23038) in ~16 hr of incubation, then washed and stained using an Affymetrix 450 Fluidics Station according to Affymetrix protocols. Finally, arrays were scanned on Affymetrix GSC3000 7G Scanner. Microarray data preprocessing was performed using apt-probeset-summarize from the Array Power Tool (APT) suite (v2.11.3) with the gc-sst-rma-sketch standard method, and the resulting expression values were log-transformed. Microarray probes targeting polymorphic regions in the BXD population were ignored in the process. For probesets targeting a same transcript, only the probeset with the highest value was considered.

**Table 1.** Gene signatures of mouse and human IBDs.

| Disease | Source | DEGs | Thresholds | Species | Tissues |
|---|---|---|---|---|---|
| DSS-induced UC | GSE131032; *Czarnewski et al., 2019* | Author-provided DEGs | Absolute $\log_2$ FC > 1 and BH-adjusted p-value<0.05 | Mouse | Colon |
| Crohn's disease and UC | GSE16879; *Arijs et al., 2009*; *Li et al., 2019* | Author-provided DEGs | Absolute FC > 1 and BH-adjusted p-value<0.01 | Human | Colon |
| Crohn's disease and UC | GSE83687; *Peters et al., 2017* | DEGs computed by limma package; *Ritchie et al., 2015* | Absolute $\log_2$ FC > 1 and BH-adjusted p-value<0.001 | Human | Colon |
| Crohn's disease | SCP1884; *Kong et al., 2023* | Author-provided DEGs | DE coefficient > 1 and BH-adjusted p-value<0.05 | Human | Colon |

BH, Benjamini–Hochberg; DEGs, differentially expressed genes; DSS, dextran sulfate sodium; FC, fold change; IBD, inflammatory bowel diseases; UC, ulcerative colitis.

## Differential gene expression analysis

General differences in mRNA expression profiles between diets were assessed using PCA. Differential expression of individual transcripts between diets was assessed using the limma R Bioconductor package (version 3.48.3) (*Ritchie et al., 2015*). Briefly, statistical significance was assessed using an empirical Bayes method (eBayes function) with an additive linear model accounting for diet and strain effect and adjusted p-values were calculated by the BH approach. Transcripts showing BH-adjusted p-value<0.05 and absolute $\log_2$ (fold change) > 0.5 were considered significantly associated with the effect of the diet.

## GSEA and ORA

Gene sets used in GSEA and ORA consisted of two parts: (1) the gene sets from the GO, KEGG, Hallmark, and Reactome databases were retrieved through the msigdbr R package (version 7.2.1) (*Liberzon et al., 2011*). (2) The gene signatures of mouse and human IBD were used as custom gene sets (*Table 1*).

GSEA was performed using clusterProfiler R package (version 3.10.1) (*Yu et al., 2012*) based on the $\log_2$(fold change) ranking using parameters (nPerm = 100000, minGSSize = 30, maxGSSize = 5000, pvalueCutoff = 1). The gene sets with absolute NES > 1 and BH-adjusted p-value<0.05 were identified as the significantly enriched gene sets.

ORA was also performed using clusterProfiler R package (version 3.10.1) (*Yu et al., 2012*) using parameters (minGSSize = 30, maxGSSize = 800). The gene sets with adjusted p-value calculated by BH < 0.05 were identified as the significantly enriched gene sets.

## Weighted gene correlation network analysis (WGCNA)

We used *WGCNA* R package (version 1.51) (*Langfelder and Horvath, 2008*) to construct co-expression networks under CD and HFD, respectively. Firstly, the correlations between all pairs of gene across all BXDs fed with CD or HFD were calculated by Pearson correlation. Then, a best soft-thresholding power of 4 and 3 was chosen using pickSoftThreshold function with parameters (*networkType* = 'signed hybrid,' blockSize = 25,000, *corFnc* = 'bicor') for CD and HFD datasets in BXD colons separately. According to the calculated correlation coefficients, a network was constructed using parameters (*networkType* = 'signed hybrid,' minModuleSize = 30, reassignThreshold = 1e-6, mergeCutHeight = 0.15, maxBlockSize = 25,000). The constructed co-expression gene modules were assigned color names, and the module eigengenes were also identified for further analyses. To detect the preserved CD-modules in the co-expression modules under HFD, we defined gene modules under CD as custom genesets and performed ORA on each HFD modules.

## TF enrichment analysis

We first constructed a lognormal background distribution using the sequences of +5 kb region around the transcription starting site (TSS) of all genes and then downloaded the mouse HOCOMOCO-v10 (*Kulakovskiy et al., 2018*) motifs from R package motifDB to perform TF enrichment analyses using R package *PWMenrich*. The significantly enriched motifs (p-value<0.001) were selected and then ranked based on the percentage of enriched promoters.

## ModQTL mapping in the BXDs

We first downloaded genotype information of each BXD mice from GeneNetwork (see here) and generated the kinship matrix of BXD mice using the leave-one-chromosome-out (LOCO) method. We then used the eigengenes of each module as phenotype input to perform ModQTL with the R package *qtl2* (version 0.28) (*Broman et al., 2019*), and the threshold of each QTL mapping analysis was obtained from a permutation test with 10,000 repeats. The peaks of QTL were calculated by *find_peaks* function with parameter: *prob = 0.95*.

The same methods were also applied to gene expression QTL mapping (eQTL), and the significance threshold of each gene was obtained from a permutation test with 1000 repeats. The significant peaks overlapped with the location of their corresponding gene were identified as *cis*-eQTL.

## Literature mining

To explore the inflammation-related genes, we first used candidate gene names and keywords ('IBD,' 'inflammatory bowel disease,' 'Ulcerative colitis,' 'Inflammation,' 'Inflammatory,' 'Crohn's disease') to search the title or abstract of associated literature using R package easyPubMed (version 2.13). Then, the genes involved in inflammation were confirmed by manual curation.

## GWAS in UKBB

We are allowed to use the UK Biobank Resource under Application Number 48020. The phenotype data of inflamed UC (Data-Field 131629, n = 6459) and CDs (Data-Field 131627, n = 3358) were firstly downloaded from UKBB (*Bycroft et al., 2018*). A total of 200,030 individuals with WGS (*Halldorsson et al., 2022*) in UK Biobank were selected and then the population of European descent (including with 1173 patients with CDs and 2295 patients with UC) was extracted for further GWAS analyses. Control individuals (n = 143,194) were included based on the following criteria: (1) individuals without noninflamed colitis (Data-Field 131631), CDs, and UC. (2) Individuals not taking any IBD-related medicine (*Supplementary file 7*).

WGS data provided by UK Biobank and used for GWAS were processed starting from pVCF files. We used REGENIE step1 to estimate population structure and then REGENIE step2 were applied to test associations between phenotypes and genetic variants and also included the following covariates in our model: the first 10 genetic PCs, age, sex, age:sex interaction, body mass index (BMI), and smoking status. All data preparation and GWAS steps were run on DNAnexus.

## Mendelian randomization (MR) analysis

eQTLs in sigmoid colon and transverse colon were selected in and their effect sizes obtained from the GTEx Portal on March 28, 2023 (v8, https://www.gtexportal.org/home/datasets, dbGaP Accession phs000424.v8.p2) (*Lonsdale et al., 2013*). No eQTLs were found for *EPHA6* but 147 and 87 eQTLs were found for *MUC4* in the sigmoid colon and transverse colon, respectively.

GWAS summary statistics for outcomes of interest, namely IBD, UC, and CDs, were obtained from the IEU OpenGWAS project (*Elsworth et al., 2020*), using the ieugwasr package (version 0.1.5, *Hemani, 2022*). As multiple GWAS exist for these diseases, we selected a representative one for each, prioritizing those with greater sample sizes and more cases, while still providing enough genetic variants for a large overlap with GTEx data (>1M SNPs). The selected GWAS were IBD (ieu-a-31) (*Liu et al., 2015*), UC (ieu-a-32) (*Liu et al., 2015*), and noncancer illness code, self-reported: CDs (ukb-b-8210) (*Elsworth et al., 2020*).

For each outcome, *MUC4* eQTLs that were present in the outcome GWAS were pruned for independence (more than 10 kbp away or r² < 0.01) using Plink v1.90b6.21 (*Purcell et al., 2007*) with the GTEx LD reference panel. In most cases, this resulted in only a single eQTL being retained, with the exception of the sigmoid colon–CDs combination, which resulted in two independent eQTLs.

MR was performed using the TwoSampleMR R package (version 0.5.6) (*Hemani et al., 2017*; *Hemani et al., 2018*). In the case with more than one eQTL, we used inverse-variance weighted MR, otherwise the Wald ratio. Because the magnitude of the normalized effect sizes provided by GTEx has no direct biological interpretation (https://gtexportal.org/home/faq#interpretEffectSize), the resulting causal effect estimates do not have an associated unit and cannot be translated into direct biological consequences. The direction (sign) of the effect remains interpretable.

## Acknowledgements

We thank the Schoonjans' and Auwerx's lab members for technical assistance and discussions and Giacomo von Alvensleben for providing the GWAS analysis pipeline in human UKBB. The work in the JA laboratory was supported by grants from the Ecole Polytechnique Fédérale de Lausanne (EPFL), the European Research Council (ERC-AdG-787702), the Swiss National Science Foundation (SNSF 31003A_179435), and the Global Research Laboratory (GRL) National Research Foundation of Korea (NRF 2017K1A1A2013124). XL was supported by the China Scholarship Council (201906050019).

# Additional information

## Funding

| Funder | Grant reference number | Author |
| --- | --- | --- |
| European Research Council | ERC-AdG-787702 | Johan Auwerx |
| National Research Foundation of Korea | NRF 2017K1A1A2013124 | Johan Auwerx |
| China Scholarship Council | 201906050019 | Xiaoxu Li |

The funders had no role in study design, data collection and interpretation, or the decision to submit the work for publication.

## Author contributions
Xiaoxu Li, Conceptualization, Resources, Data curation, Validation, Investigation, Visualization, Methodology, Writing - original draft, Writing – review and editing; Jean-David Morel, Giorgia Benegiamo, Writing – review and editing; Johanne Poisson, Jonathan Sulc, Methodology; Alexis Bachmann, Evan Williams, Resources; Alexis Rapin, Data curation; Alessia Perino, Conceptualization, Writing – review and editing; Kristina Schoonjans, Maroun Bou Sleiman, Johan Auwerx, Supervision, Writing – review and editing

## Author ORCIDs
Xiaoxu Li http://orcid.org/0000-0001-5121-9190
Giorgia Benegiamo http://orcid.org/0000-0001-7164-6771
Johanne Poisson https://orcid.org/0000-0002-0183-845X
Evan Williams https://orcid.org/0000-0002-9746-376X
Kristina Schoonjans http://orcid.org/0000-0003-1247-4265
Johan Auwerx http://orcid.org/0000-0002-5065-5393

## Ethics
Human subjects: We are allowed to use the UK Biobank Resource under Application Number 48020. All animal procedures were approved by the veterinary office of Canton Vaud under animal experimentation license number VD2257.

Reviewer #1 (Public Review): https://doi.org/10.7554/eLife.87569.3.sa1
Reviewer #2 (Public Review): https://doi.org/10.7554/eLife.87569.3.sa2
Author Response: https://doi.org/10.7554/eLife.87569.3.sa3

# Additional files

## Supplementary files
• Supplementary file 1. Co-expression networks under chow diet (CD) or high-fat diet (HFD). Related to *Figure 3*.

• Supplementary file 2. Genes under quantitative trait locus (QTL) peak of module HFD_M28. Related to *Figure 5*.

• Supplementary file 3. Associations between genetic variants of EPHA6 and inflammatory bowel disease (IBD). Related to *Figure 5* and its *Figure 5—figure supplement 1*.

• Supplementary file 4. Associations between genetic variants of MUC4 and inflammatory bowel disease (IBD). Related to *Figure 5* and its *Figure 5—figure supplement 1*.

• Supplementary file 5. G-MAD result for *Epha6* in mice and *MUC4* in humans. Related to *Figure 5*

• Supplementary file 6. Mendelian randomization (MR) result for MUC4.

• Supplementary file 7. Medicine for human inflammatory bowel disease (IBD).

• MDAR checklist

## Data availability

The data that support the findings and code used to perform analyses are freely on GitHub (copy archived at *Li, 2023*). The microarray data have been deposited in GEO under accession code GSE225791.

The following dataset was generated:

| Author(s) | Year | Dataset title | Dataset URL | Database and Identifier |
|---|---|---|---|---|
| Auwerx J, Li X, Bachmann A, Rapin A, Bon Sleiman M, Morel JD | 2023 | Genetic and dietary modulators of the inflammatory response in the gastro-intestinal tract of the BXD mouse genetic reference population | https://www.ncbi.nlm.nih.gov/geo/query/acc.cgi?acc=GSE225791 | NCBI Gene Expression Omnibus, GSE225791 |

The following previously published datasets were used:

| Author(s) | Year | Dataset title | Dataset URL | Database and Identifier |
|---|---|---|---|---|
| Czarnewski P, Villablanca EJ | 2019 | Time-series reveals processes underlying colon inflammation and repair | https://www.ncbi.nlm.nih.gov/geo/query/acc.cgi?acc=GSE131032 | NCBI Gene Expression Omnibus, GSE131032 |
| Arijs I, Van Lommel L, Van Steen K, De Hertogh G, Geboes K, Schuit F, Rutgeerts P | 2009 | Mucosal expression profiling in patients with inflammatory bowel disease before and after first infliximab treatment | https://www.ncbi.nlm.nih.gov/geo/query/acc.cgi?acc=GSE16879 | NCBI Gene Expression Omnibus, GSE16879 |
| Peters LA, Perrigoue J, Mortha A, Iuga A | 2017 | A functional genomics predictive network model identifies regulators of inflammatory bowel disease: Mount Sinai Hospital (MSH) Population Specimen Collection and Profiling of Inflammatory Bowel Disease | https://www.ncbi.nlm.nih.gov/geo/query/acc.cgi?acc=GSE83687 | NCBI Gene Expression Omnibus, GSE83687 |
| Kong L, Pokatayev V, Lefkovith A, Carter GT, Creasey EA, Krishna C, Subramanian S, Kochar B, Ashenberg O, Lau H, Ananthakrishnan AN, Graham DB, Deguine J, Xavier RJ | 2023 | The landscape of immune dysregulation in Crohn's disease revealed through single-cell transcriptomic profiling in the ileum and colon | https://singlecell.broadinstitute.org/single_cell/study/SCP1884/ | Broad Single Cell Portal, SCP1884 |

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
