## [Editor Report · eLife assessment]

This **fundamental** study provides a framework for leveraging systems genetics data to dissect mechanisms of gut physiology. The authors provide **compelling** analyses to highlight diverse modes of interrogating intestinal inflammation, dietary response, and consequent impacts on inflammatory bowel disease. As a resource, it will have great utility for linking genetic variation and diet to gut-related pathophysiologies.

---

## [Referee Report · Reviewer #1 (Public Review)]

Using the colon transcriptomes of 52 BXD mouse strains fed either chow or a high-fat diet (HFD), Li et al. present their findings on gene-by-environment interactions underpinning inflammation and inflammatory bowel disease (IBD). They discovered modules that are enriched for IBD-dysregulated genes using co-expression gene networks. They determined Muc4 and Epha6 to be the leading candidates causing variations in HFD-driven intestinal inflammation by using systems genetics in the mouse and integration with external human datasets. In their analysis, they concluded that their strategy "enabled the prioritization of modulators of IBD susceptibility that were generalizable to the human situation and may have clinical value." This dataset is intriguing and generates hypotheses that will be investigated in the future. However, there were no mechanistic or causation-focused investigations; the results were primarily observational and correlative.

---

## [Referee Report · Reviewer #2 (Public Review)]

In this paper, the authors seek to identify genes that contribute to gut inflammation by capitalizing on deep phenotyping data in a mouse genetic reference population fed a high-fat or chow diet and then integrating it with human genetic data on gut inflammatory diseases, such as inflammatory bowel disease (IBD) and Ulcerative Colitis (UC). To achieve this the authors performed genome-wide gene expression in the colon of 52 BXD strains of mice fed either a high-fat or chow diet. From this analysis, they observed significant variation in gene expression related to inflammation among the 52 BXD strains and differential gene expression of inflammatory genes fed a high-fat diet. Overlaying this data with existing mouse and human data of inflammatory gut disease identified a significant enrichment. Using the 52 BXD strains the authors were able to identify specific subsets of strains that were susceptible and resistant to gut inflammation and analysis of gene expression within the colon of these strains was enriched with mouse and human IBD. Furthermore, analysis of cytokine levels of IL-10 and IL-15 were analyzed and found to be increased in resistant BXD strains and increased in susceptible BXD strains.

Using the colon genome-wide gene expression data from the 52 BXD strains, the authors performed gene co-expression analysis and were able to find distinct modules (clusters) of genes that correlated with mouse UC and human IBD datasets. Using the two modules, termed HFD_M28 and HFD_M9 that correlated with mouse UC and human IBD, the authors performed biological interrogation along with transcription factor binding motif analysis to identify possible transcriptional regulators of the module. Next, they performed module QTL analysis to identify potential genetic regulators of the two modules and identified a genome-wide significant QTL for the HFD_M28 on mouse chromosome 16. This QTL contained 552 protein-coding genes and through a deduction method, 27 genes were prioritized. These 27 genes were then integrated with human genetic data on IBD and two candidate genes, EPHA6 and MUC4 were prioritized.

Overall, this paper provides a framework and elegant use of data from a mouse genetic reference population coupled with human data to identify two strong candidate genes that contribute to human IBD and UC diseases. In the future, it will be interesting to perform targeted studies with EPHA6 and MUC4 and understand their role in gut inflammatory diseases.

---

## [Author Response]

The following is the authors’ response to the original reviews.

**Reviewer #1 (Recommendations For The Authors):**
- There were no mechanistic or causation-focused investigations that could have greatly strengthened the study. The study is ultimately providing two prioritized candidate genes that may be causative, reactive, or independent of the disease.

We thank the reviewer for their positive assessment and agree that our study lacks formal causal analyses. We are aware of this limitation and have made it clear throughout the text. Through triangulation of evidence across tissues and species, we point to very interesting candidates that merit further study, which is the usual scope of such systems genetics investigations. Nevertheless, to introduce some causal inference and reinforce the human relevance of our results, we have performed Mendelian randomization (MR) analysis to investigate the potential associations between *MUC4*’s gene expression in human colons and the risk of IBD. *EPHA6* lacks detectable eQTLs in human colon so we could not include it in this analysis. We found suggestive evidence that increased expression of *MUC4* in the sigmoid, but not transverse, colon may increase the risk of IBD (nominal p = 0.033).

The description in the manuscript:

However, it is unclear through what mechanisms the genetic variants in the candidate genes affect IBD susceptibility. One possibility is that genetic variation leads to altered levels of expression of the gene, ultimately affecting disease susceptibility. To test this possibility, we examined the GTEx resource (GTEx Consortium, 2013) and found that *MUC4*, but not *EPHA6*, has cis-eQTLs in the sigmoid and transverse colon. To establish likely causal links with IBD incidence, we used these associations as instruments in a two-sample Mendelian randomization (MR) (Hemani, Tilling and Smith, 2017; Hemani *et al.*, 2018) analysis. Using publicly available GWAS summary statistics for IBD, Crohn’s disease, and ulcerative colitis (Liu *et al.*, 2015; Elsworth *et al.*, 2020) as outcomes, we found suggestive evidence that increased expression of *MUC4* in the sigmoid, but not transverse, colon may increase the risk of IBD (nominal P value = 0.033, Appendix 1 - Table 6). No eQTLs were reported for *EPHA6* in the colon, precluding us from investigating the potential consequences of changes in its expression in these tissues.

- Figures 3 and its supplement Figure 1: Among the 39 modules, the authors have only focused on significantly overlapping up-regulated IBD-related gene modules in both CD (M28 and M32) and HFD (M9 and M28) for their follow up analyses in Figures 4 and 5 to prioritize candidate genes. However, this reviewer thinks there is great value in also focusing on significantly overlapping down-regulated IBD-related gene modules in both CD (M17) and HFD (M15 and M26) for their follow up candidate gene prioritization analyses.

Thank you for your suggestion. We had initially performed overrepresentation analyses in HFD_M15, HFD_M26 and CD_M17, but did not find enrichments related to inflammation (see Author response image 1 below). We did not include this result in the manuscript.

**Author response image 1. sa3fig1:** Dot plot showing the enrichment of IBD-related modules in hallmark genesets. Gene ratios higher than 0.1 are shown and represented by dot size. Dots are colored by -Log10(BH-adjusted P values).

We also checked the module QTL mapping for the significantly overlapping down-regulated IBD-related gene modules in both CD and HFD. We did not find any loci that are significantly associated with these modules, indicating that they are not modulated by genetic variation and hence are less likely to inform on IBD susceptibility.

The description in the manuscript:

The ModQTL analysis was also performed on the modules that are significantly enriched in IBD-downregulated genes (HFD_M15, HFD_M24, and HFD_M26), but no significant or suggestive QTLs were detected. Therefore, we focused on the QTL for IBD-induced genes in HFD_M28 and annotated its candidate genes based on three criteria (Figure 5B).

**Reviewer #2 (Recommendations For The Authors):**

- One small addition that would be nice would be to indicate if the two candidate genes have cis eQTL in human tissues and/or have any protein-coding variants in humans. This would provide nice additional evidence of causality for these two genes.

Thank you for your positive assessment and suggestion. *MUC4* and *EPHA6* both have protein-coding variants in humans that were listed in the Appendix – Table 3 and Table 4. In addition, cis-eQTLs have been found for *MUC4* in both the sigmoid and transverse colon in humans (GTEx, https://gtexportal.org/home/locusBrowserPage/ENSG00000145113.21). As indicated in our response to the first comment of Reviewer #1, we have now performed mendelian randomization on human eQTL for *MUC4*. However, no eQTLs were reported for *EPHA6* in the colon, preventing us from performing MR analysis on its expression.

- Also, it would be helpful to include the size of the modules in the text of the manuscript. Especially the two modules that were followed up on.

Thank you for your suggestion, we have indicated the size of IBD-related modules in the text of the manuscript.

The description in the manuscript:

Enrichment analyses indicated that modules HFD_M9 (484 genes), HFD_M16 (328 genes), and HFD_M28 (123 genes) were enriched with genes that are upregulated by DSS-induced colitis, while HFD_M15 (368 genes), HFD_M24 (159 genes), and HFD_M26 (135 genes) were significantly enriched with downregulated genes **(**Figure 3C**).** Of note, more than 20% of genes involved in HFD_M9 and HFD_M28 were part of the dysregulated genes of the acute phase of mouse UC (day6 and day7) (Figure 3C). Interestingly, genes perturbed during IBD pathogenesis in humans were also enriched in HFD_M9 and HFD_M28 (Figure 3C).

While IBD-related genes were predominantly found in HFD modules, we also found that two modules, CD_M28 (185 genes) and CD_M32 (142 genes), in CD-fed mouse colons were associated with IBD (Figure 3—figure supplement 1A**)**. These two-modules significantly overlapped with the IBD-related HFD_M9 and HFD_M28 modules, respectively (BH-adjusted P value < 0.05) (Figure 3—figure supplement 1B**)**. Moreover, the molecular signatures underlying human UC and Crohn’s disease were also clustered in these two modules (CD_M28 and CD_M32) under CD (Figure 3—figure supplement 1C). Collectively, the co-expression and enrichment analyses identify HFD_M9 and HFD_M28 as IBD-related modules on which we focus our subsequent investigation.